# Four millennia of long-term individual foraging site fidelity in a highly migratory marine predator

Eric. J. Guiry ® [1,2,3✉], Margaretta James[4], Christina Cheung[5] & Thomas C. A. Royle[6]

Theory and field studies suggest that long-term individual foraging site fidelity (IFSF) may be an important adaptation to competition from increasing population. However, the driving mechanisms and extent of long-term IFSF in wild populations of long-lived, migratory animals has been logistically difficult to study, with only a few confirmed instances. Temporal isotopic datasets can reveal long-term patterns in geographical foraging behaviour. We investigate the isotopic compositions of endangered short-tailed albatross (*Phoebastria albatrus*) over four millennia leading up to their near-extinction. Although not exhibited by short-tailed albatross today, we show past sub-populations displayed a high-degree of long-term IFSF, focusing on the same locations for hundreds of generations. This is the first large-scale evidence for the deep antiquity of long-term IFSF and suggests that it's density-driven. Globally, as populations of species like short-tailed albatross continue to recover from overexploitation, potential for resurgence of geographic specialization may increase exposure to localized hazards, requiring closer conservation monitoring.

[1] School of Archaeology and Ancient History, University of Leicester, Mayor's Walk, Leicester LE1 7RH, UK. [2] Department of Anthropology, Trent University, 1600 West Bank Drive, Peterborough, ON K9L 0G2, Canada. [3] Department of Anthropology, University of British Columbia, 6306 NW Marine Drive, Vancouver, BC V6T 1Z1, Canada. [4] Land of Maquinna Cultural Society, Mowachaht/Muchalaht First Nation, Tsaxana (Gold River), BC V0P 1G0, Canada. [5] Research Unit, Analytical, Environmental & Geo-Chemistry, Department of Chemistry, Vrije Universiteit Brussel, AMGC-WE-VUB, Pleinlaan 2, 1050 Brussels, Belgium. [6] Ancient DNA Laboratory, Department of Archaeology, Simon Fraser University, 8888 University Drive, Burnaby, BC V5A 1S6, Canada. ✉email: eguiry@lakeheadu.ca

Understanding variation in foraging mobility behaviour is critical for developing effective conservation and restoration programs for seabirds and other migratory taxa that are recovering from overexploitation[1,2]. The short-tailed albatross (*Phoebastria albatrus*), a large North Pacific seabird, is currently undergoing a population recovery. Once numbering in the millions, the species was brought to the brink of extinction by feather hunters between the 1880s and 1930s, which left no functioning breeding colonies intact[3,4]. While still classified as Vulnerable by the IUCN[5] and Japan[6], Threatened by Canada[7,8], and Endangered by the United States[9–11], careful conservation efforts aimed at monitoring and protecting these birds both at breeding colonies (for reviews see refs. [10,11]) and across much of their foraging range (e.g.,[7,10]) have resulted in exponential population growth over recent decades[5,7]. Despite these gains, the species remains at less than 1% of its pre-collapse population levels and continued progress will be dependent, in part, on our ability to anticipate evolving relationships between potential threats (e.g., fisheries by-catch) and behavioural patterns of short-tailed albatross that may be shifting in response to population growth[12–14].

Short-tailed albatross forage widely over both neritic (coastal) and oceanic (offshore) habitats of the North Pacific (from the Philippine Sea to Mexico; Fig. 1), but are known to concentrate foraging in hotspots associated with bathymetric (shelf breaks: steep bottom topographies that generate strong vertical mixing) and climatic variables favouring greater abundances of nutrient upwelling and stronger winds[12,15,16]. Tracking studies and sighting records have identified ontogenetic shifts in foraging mobility behaviour between juvenile and adult birds. In general, juveniles explore a wider range of the North Pacific with greater habitat diversity use, including oceanic, shelf, and shelf-break areas. In contrast, as juveniles age, foraging range size decreases as birds begin to spend a larger proportion of their time on shelf-break areas with higher productivity[12,16,17]. Presently, in some areas of their range, potential threats like fisheries by-catch are mitigated by taking preventative actions (e.g., using seabird deterrents) in known foraging hotspots (e.g.,[14]). As population growth continues, understanding the factors governing how and where short-tailed albatrosses select, and apportion their time between, specific foraging areas may be key for developing effective conservation approaches[18–20].

In addition to understanding habitat and location preferences, another potentially critical element for anticipating vulnerable areas in the short-tailed albatross' foraging range is individual foraging site fidelity (IFSF; when birds choose to forage primarily in a subset of the region used by the entire population). When segments of a population exhibit a high degree of long-term IFSF (i.e., IFSF patterns repeated over multiple annual cycles), it can leave them vulnerable to hazards that are concentrated in relatively small, localised areas[21,22]. While long-term IFSF is increasingly recognised in seabird populations[23–25], its prevalence among short-tailed albatross remains unknown. Limited multi-year observations of juveniles suggest that a small number of individuals may be beginning to develop towards preferences for repeated use of the same areas, though, on the whole, most individuals show low levels of spatial fidelity[12].

While modern observations and tracking data are helpful, low population sizes means that behaviours influenced by carrying capacity and population density are likely to change as the species recovers. For instance, observed overlap between feeding habitats of adults and juvenile short-tailed albatross[12] breaks with expectations based on observations in other albatross species, which show low overlap between foraging areas of older and younger individuals (e.g.,[26,27]). While this may be related to a variety of processes, density-driven factors like a probable absence of competitive exclusion or competition avoidance between adult and juvenal short-tailed albatross are likely candidates[12]. Such uncertainties highlight how use of data from modern populations, while a key source of information for monitoring and predicting behaviour, may be unable to address some important questions about species behaviour as populations continue to rebound.

As populations grow it is likely that we can, for instance (among many potential responses), expect the present distribution and use intensity of short-tailed albatross foraging hotspots to expand[12]. This is because increasing competition in current hotspots could create incentives for some birds to shift or intensify their focus on other areas, potentially creating new foraging hotspots. Archaeological and historical observations can give us a general sense of past distributions and abundance that, in the context of modern data on probable prime habitat areas (e.g., bathymetry and prevailing meteorological trends), can help to predict areas that may be good candidates for increased short-tailed albatross foraging activity in the future (e.g.,[14]). But these sources of data alone cannot help anticipate the development of fidelity for particular migratory and area-specific foraging behaviours at the individual or population level. To explore patterns in

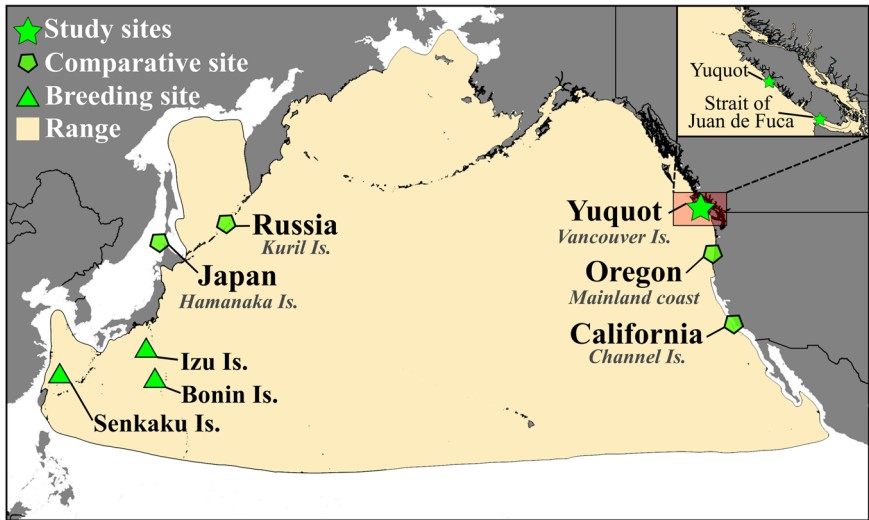

**Fig. 1 Map of study region showing locations.** Map shows breeding islands and archaeological and historical sampling sites in the context of the North Pacific Ocean and the short-tailed albatross' foraging range. Range data are from BirdLife International[5].

long-term IFSF, we can look to biomolecular archives preserved in archaeological and historical specimens for evidence of the extent to which fidelity to specific foraging areas occurred in the past.

In this study, we use isotopic analyses of bone collagen from 95 archaeological short-tailed albatross specimens to explore long-term (multi-year) patterns in foraging characteristics of birds harvested over a 4250-year period near a single location, the Yuquot site (DjSp-1), located on the west coast of Vancouver Island (British Columbia [BC], Canada, Fig. 1), at the far edge of species' foraging range. Analysis also included feathers from two nineteenth-century specimens collected in the nearby Strait of Juan de Fuca (Fig. 1). The large sample and temporal span, restricted geographical sampling area, and distant location (relative to the breeding sites, located thousands of km to the west) represented by this dataset provides an ideal context in which to explore variation in relationships between fidelity for foraging areas used by short-tailed albatrosses for millennia prior to their near-extinction collapse in the 1930s.

**Short-tailed Albatross foraging ecology and conservation**. Due to the severe population reduction following the species' collapse, as well as its large foraging range, studies of short-tailed albatross mobility and diet have been limited[3,4]. Today, short-tailed albatross breed on only a few islands, primarily in the far Western Pacific Ocean (mainly on Izu [Tori-Shima], Senkaku [Minami-kojima], and Bonin [Mukojima] Island groups; Fig. 1), though several other islands in the same region were historically used for breeding[3,11]. Though long-extirpated, there is also some evidence that the species' breeding distribution included islands in the Atlantic Ocean during the Middle Pleistocene (c. 405,000 years BP[28]) and some have further hypothesised the past existence of a more recent but now-extinct Eastern Pacific breeding colony in the Aleutian Islands[29] (Alaska, USA); however, at present, archaeological and DNA evidence do not support this[30]. Tracking studies and sighting records have shown that, today, while present across their range, short-tailed albatross (particularly adults and maturing juveniles) spend more time foraging in the Western (nearer to their breeding islands) and Northern Pacific, and focus comparatively less time along the more distant Eastern Pacific areas of their range, including the coastal regions of Canada, the continental United States, and Mexico[12]. These studies also show that most short-tailed albatross move to forage seasonally across different areas of their range. While diet studies are rare[17], it is widely accepted that short-tailed albatross are generalist foragers targeting foods available at the ocean surface, including squid, fish, and carrion[11].

Short-tailed albatross are often observed foraging near fishing operations where they take advantage of discard and offal (e.g.,[14]). With respect to the species' protection across its foraging range, this behaviour represents a potential threat as short-tailed albatross interacting with fishing fleets and their gear can result in injury or death for some birds (e.g.,[18,19,31]). Significant efforts have been invested in understanding overlap between short-tailed albatross foraging hotspots and fisheries activities in order to monitor and mitigate hazards for foraging birds[13–16,18]. In this context, understanding the potential for, and nature of, shifting behaviour in the future may be key to ensuring the continued success of efforts to rehabilitate the species.

**Stable isotope theory and interpretive context**. Stable carbon ($\delta^{13}$C) and nitrogen ($\delta^{15}$N) isotope compositions of archaeological and museum archived animal tissues can provide powerful indicators for exploring mobility and dietary patterns in past and present eco-systems (e.g.,[32,33]). Isotopic compositions of consumer tissues reflect

the foods they consume during the period over which the tissue was forming and remodelling[34]. In contrast to most other tissues (which turn over on a scale of days, weeks, or months), isotopic compositions from bone collagen, which remodels slowly over the entire lifespan on an individual, reflect an average of foods consumed over the last several years of an individual's life[34–36]. For this reason, isotopic variation in bone collagen among individual birds can be used to indicate temporally broad-scale (multi-annual) differences in food sources (e.g.,[37]). The long-term perspective on individual behaviour provided by bone collagen isotopic compositions, therefore, offers an ideal approach for exploring spatiotemporal relationships between life-time scale patterns in foraging mobility in short-tailed albatross populations represented by archaeological collections in different regions of the species' foraging range.

A wide range of variables influence the $\delta^{13}$C and $\delta^{15}$N of marine consumers. For instance, $\delta^{15}$N values of consumer tissues increase systematically with each trophic level[38] and are therefore typically used to explore trophic relationships. Stable carbon isotope compositions, in contrast, are passed between prey and consumers with comparatively little change[39] and are often used to trace energy pathways between consumers and primary producers[40]. Beyond reconstructing diet, a wide range of processes governing carbon and nitrogen sources and cycling in aquatic environments can result in spatial and temporal variation in isotopic baselines (for reviews see refs. [41,42]), which can provide a framework for tracking the movement of animals between isotopically distinct regions. For instance, key (and often interrelated) variables that govern isotopic compositions of primary producers in oceans include sea surface temperature and $CO_2$ concentrations, productivity and nutrient limitation, phytoplankton physiology and growth rates, prevalence of nitrogen fixation or denitrification, and currents causing mixing between regions with differing isotopic baselines[41,43–45]. These variables are responsible for creating broad regional isotopic baseline differences among food webs across the short-tailed albatross's foraging range[46], such that groups of individuals that consistently forage among specific, differing suites of regions can have distinctive isotopic niches. For long-term retrospective studies such as this, it is important bear in mind that processes governing baseline isotopic variation can vary over time in response to changing climatic (e.g.,[47,48]) and anthropogenic impacts (e.g.,[49–51]). While this kind of temporal variation will always remain a source of potential interpretive uncertainty for retrospective studies, it will be of less concern in research contexts where isotopic variation associated with temporal shifts is likely smaller than that arising from the dietary or migratory behavioural patterns of interest.

Previous isotopic analyses provide a framework for exploring isotopic variation in ancient short-tailed albatross bone collagen from birds recovered from the Yuquot site (Fig. 1), including samples from California ($n = 49$ from unspecified sites on the Channel Islands dating to ca. 500–1500 CE), Japan ($n = 46$ from the Hamanaka 2 site and $n = 10$ from unspecified sites in the Hamanaka Islands dating to 600–1100 CE), Oregon ($n = 17$ from unspecified sites dating to ca. 500–1500 CE), and Russia ($n = 35$ from unspecified sites on the Kuril Islands dating ca. 500–1500 CE)[30,46]. In addition to isotopic analyses of 'bulk' bone collagen (the approach we use here), a small subset of these previous analyses have also explored isotopic variation (California, $n = 17$; Russia, $n = 11$; Japan, $n = 9$) among the individual amino acids that compose bone collagen[46]. By comparing isotopic compositions among amino acids (AAs) that are expected to be isotopically altered during metabolism and protein synthesis with those that are not (i.e., essential AAs), these analyses provided an opportunity to assess the relative impact of variation in individual diets (e.g., types of prey, trophic level) vs. feeding in

areas with differing isotopic baselines (e.g.,[52,53]). Vokhshoori and colleagues[46] found that the dominant driver of isotopic variation in short-tailed albatross bone collagen was foraging behaviour linked with isotopically distinctive geographical regions used by different individuals. In this context, variation in diet (e.g., trophic level or the type of foods consumed) was shown to be a less impactful, secondary factor determining short-tailed albatross isotopic compositions. Another finding of relevance is that patterns in isotopic niche size[54] calculated using both 'bulk' bone collagen and individual AAs were similar, suggesting that both provide a useful approach form exploring isotopic patterns associated with variation in foraging locations. In that context, taken as a whole, the body of previously published 'bulk' bone collagen (hereafter, simply 'bone collagen') data provides a series of geographically varied comparative contexts in which to assess variation in isotopic niches associated with foraging in differing habitats used by short-tailed albatross harvested at the Yuquot site.

## Results

Ninety-eight percent ($n = 93$ of 95; Supplementary Data 1) of bone collagen samples have elemental concentrations[55] and C:N ratios passing quality control (QC) criteria (liberal criteria used[56]). A further comparison of $\delta^{13}C$ and C:N showed no relationship ($n = 93$, Pearson's $r$ –0.051, $p = 0.628$), providing additional indication that isotopic compositions have not been altered by contamination[56]. Mean isotopic compositions from all four types of bone samples (Tarsometatarsi, humeri, coracoids, and wing digits; Supplementary Table 1) showed show little variation in $\delta^{13}C$ (range for means = 0.2‰) and $\delta^{15}N$ (range for means = 0.8‰). These differences are smaller than expected for typical intra-skeletal isotopic variation[35] indicating that bone type has not influenced isotopic patterns. Samples passing QC criteria produced mean $\delta^{13}C$ and $\delta^{15}N$ values of –14.6 ± 0.4‰ and +18.0 ± 0.7‰, respectively (Fig. 2b and Supplementary Data 1). Broken down by temporal zone (Table 1 and Fig. 2b1 and b2),

group means show little variation ($\delta^{13}C$ range = 0.18‰; $\delta^{15}N$ range = 0.82‰) and there are no statistically significant differences between sequential time periods. Among time periods for both isotopic compositions, Shapiro–Wilk tests showed that only Zone 4 $\delta^{15}N$ was not normally distributed (i.e., $p$ = >0.05; Supplementary Table 2). Levene's tests showed that significant differences in variances occur among $\delta^{13}C$ groups ($p = 0.044$) but not among $\delta^{15}N$ groups ($p = 0.480$). A Welch's $F$-test showed no significant differences in mean $\delta^{13}C$ between time periods ($F = 0.555$, df = 18.58, $p = 0.651$). A Mann–Whitney $U$-test with a Bonferroni correction showed that among temporal group means for $\delta^{15}N$, no significant differences occur except for between Zones 4 and 2 ($U = 0.613$, $p = 0.004$; Supplementary Table 3), though the difference was small (0.82‰).

In comparison with bone collagen samples from archaeological sites in other regions of the North Pacific, short-tailed albatross harvested at the Yuquot site show much less variation (Table 2, Fig. 2a, and Supplementary Fig. 1), with SEA$_B$ and SEA$_C$ values that are approximately half the size (or less) of other groups. In the context of previously published short-tailed albatross data, showing that isotopic variation is linked more strongly with foraging geography (i.e., use of areas with differing isotopic baselines) than with diet[46], this indicates large differences in the isotopic variation and niche sizes between our sample (small) and short-tail albatross harvested in other regions of the North Pacific (all comparatively large; Fig. 2a–f). We performed a further bootstrapping simulation to assess the probability that, across 1000 iterations, a random selection of 93 data points (the size of our sample) from the entire sample (all regions, $n = 250$) would produce standard deviations of $\delta^{13}C$ and $\delta^{15}N$ larger than those observed at the Yuquot site. Results (Supplementary Fig. 2a, b) show that this occurred in only one case for $\delta^{15}N$ (or 0.01% of all simulations) and in no cases at all for $\delta^{13}C$, providing further proof of the distinctiveness of the low isotopic variation observed among short-tailed albatross harvested at Yuquot.

Short-tailed albatross feathers (with $\delta^{13}C$ corrected to bone collagen by +1.6‰[57]), dating from 1889 CE, produced isotopic compositions falling near but slightly outside the convex hull of our sample from short-tailed albatross bone collagen (increasing TA by only 0.3, from 4.24 to 4.56; Supplementary Fig. 3). We do not include these among statistical comparisons using bone collagen because isotopic compositions of feather (representing diet over a course of weeks) are not directly comparable with bone (representing an average of several years[34,36]). The coarse similarity between feather and bone collagen isotopic compositions does, however, suggest that short-tailed albatross foraging in the study region maintained foraging behaviours that were similar to those observed in our archaeological population until at least the late 1880s CE.

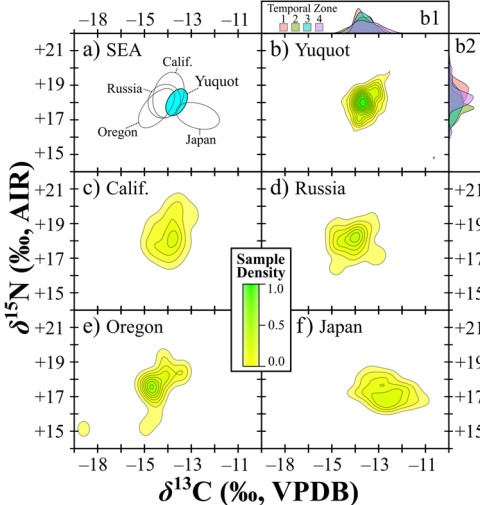

**Fig. 2 Bone collagen isotopic variation among archaeological short-tailed albatross.** Assemblages from the Yuquot site and other regions[30,46] are shown in panes: (**a**) bivariate ellipses with confidence levels set at 0.5 for $\delta^{13}C$ and $\delta^{15}N$ from the Yuquot sample (shown in blue) vs. all other groups (see Table 2 for niche sizes and $n$); density contours for $\delta^{13}C$ and $\delta^{15}N$ of samples from the Yuquot site (**b**), California (**c**), Russia (**d**), Oregon (**e**), and Japan (**f**); $\delta^{13}C$ (**b1**) and $\delta^{15}N$ (**b2**) density histograms from Yuquot temporal zones (see Table 1 for timeframes and $n$). Analytical uncertainty is shown in lower right of panel **b**.

**Table 1 Mean isotopic compositions for short-tailed albatross bone collagen from the Yuquot site shown by temporal zone.**

| Zone | Period | $n$ | $\delta^{13}C$ (‰) | $\delta^{15}N$ (‰) |
|---|---|---|---|---|
| 1 | Pre-2300 to 1000 BCE | 5 | –14.7 ± 0.3 | 18.5 ± 0.5 |
| 2 | 1000 BCE to 800 CE | 25 | –14.6 ± 0.4 | 17.7 ± 0.5 |
| 3 | 800 to 1789 CE | 39 | –14.6 ± 0.3 | 18.0 ± 0.7 |
| 4 | 1789 to 1966 CE | 24 | –14.5 ± 0.5 | 18.3 ± 0.7 |

Temporal zones are based on well-dated ($^{14}C$) stratigraphic contexts (for dates see ref.[69]). There are no statistically significant differences in mean $\delta^{13}C$ and $\delta^{15}N$ between sequential time periods. Note that, while the terminus for archaeological contexts that make up Zone 4 occurs in the mid-twentieth century, historical records suggest that collection of short-tailed albatross at the Yuquot site ceased before the twentieth century. For this reason we have not applied Suess corrections[51] to Zone 4 $\delta^{13}C$.

**Table 2 Mean isotopic compositions as well as area calculations for all locations.**

| Location | $n=$ | $\delta^{13}$C (‰) | $\delta^{15}$N (‰) | TA | SEA | SEA$_B$ | SEA$_C$ |
|---|---|---|---|---|---|---|---|
| Yuquot site | 93 | −14.6 ± 0.4 | 18.0 ± 0.7 | 4.237 | 0.763 | 0.765 | 0.771 |
| California[46] | 49 | −14.9 ± 0.5 | 18.5 ± 1.1 | 6.398 | 1.791 | 1.763 | 1.829 |
| Japan[30,46] | 56 | −13.7 ± 0.8 | 17.3 ± 0.6 | 5.515 | 1.396 | 1.385 | 1.422 |
| Oregon[46] | 17 | −15.6 ± 0.9 | 17.5 ± 1.2 | 7.723 | 2.409 | 2.435 | 2.569 |
| Russia[46] | 35 | −15.1 ± 0.6 | 18.1 ± 0.9 | 5.848 | 1.596 | 1.561 | 1.664 |

Convex hull (Total Area [TA]) and standard ellipse areas (SEA, SEA$_B$, and SEA$_C$) for archaeological short-tailed albatross bone collagen are shown for the Yuquot site and other sampling regions. For complete contextual data for samples from each region see Fig. 1 and Supplementary Data 1.

## Discussion

Our isotopic results provide new insights for understanding both seabird foraging mobility in general and short-tailed albatross behaviour in the eastern-central portion (i.e., west coast of Vancouver Island) of their range specifically. First, the much smaller isotopic niche observed for our sample, relative to other samples across the species' range, suggests that individuals from the Yuquot site shared a more narrowly defined and geographically similar foraging area than birds collected at other locations. Second, similarity between the mean isotopic compositions across sequential time periods, suggests that the foraging habitats used by short-tailed albatross harvested near the Yuquot site across our 4250-year study period were stable and underwent little change.

This isotopic and temporal pattern could result from two different foraging mobility behaviour scenarios. First, low isotopic variation among short-tailed albatross recovered from the Yuquot site could be explained by birds feeding in equal proportions in a wide variety of isotopically distinct areas across their range. This explanation seems unlikely, however, as, if this were the case, we would not expect to see differences in isotopic variation between archaeological samples from Yuquot and sites in adjacent regions (e.g., Oregon and California; Table 2 and Fig. 2a, c, e), which were also accessible to migrating short-tailed albatross. In other words, to account for the much narrower range of isotopic variation (i.e., very small isotopic niche) in short-tailed albatross harvested at Yuquot, we would need to assume that harvesting efforts were: 1) targeting only birds that used an isotopically similar suite of foraging locations across the North Pacific; 2) somehow identifying and selectively excluding individuals that used other areas; and, 3) that this occurred consistently over thousands of years. Isotopic compositions for short-tailed albatross harvested in Oregon, for instance, show a much wider range of diversity (Fig. 2a, b, e and Supplementary Fig. 1) and, given the temporal overlap and closer geographical proximity between these archaeological assemblages (Fig. 1), these birds presumably could also have accessed (and instead been harvested) in areas close to the Yuquot site. In this context, absence of the more varied isotopic compositions, which are seen in smaller samples from other regions at the eastern margins of the species' range, makes this explanation improbable.

A second, more parsimonious, explanation is that short-tailed albatross harvested near the Yuquot site shared a similarly high behavioural and geographical affinity for long-term IFSF focusing primarily on past hotspot areas near the west coast of Vancouver Island. This is because repeated use of a smaller number of foraging areas that are primarily located in regions with a shared isotopic baseline, as could be expected with long-term IFSF, would be both consistent with the smaller isotopic niche observed here and also provide a more realistic explanation for how seabird harvesting practices at the Yuquot site could have selectively targeted short-tailed albatross with such a low degree of isotopic variation (Fig. 2a).

In considering this explanation, a number of points should be borne in mind. First, it is important to point out that this interpretation does not exclude the possibility that short-tailed albatross harvested at the Yuquot site were not also potentially foraging, in a more limited capacity, across a wide variety of other areas throughout their range. It also worth noting that, while our isotopic data do not provide a quantitative measure of the foraging area size or definitive indicator of geographical locations used by these short-tailed albatross, it stands to reason that they likely focused primarily on a small number of hotspots located in closer proximity to the Yuquot site. Finally, while we use the term IFSF to describe this pattern, it is important to note that our data do not provide an indication of whether foraging site fidelity occurred among unconnected individuals or, rather, was shared among many individuals from the same colony (i.e., colony-level segregation). Bearing in mind that each scenario could have different ecological and conservation implications, we note that the distinction between them can be vague and would be extremely difficult to assess retrospectively. Because these data pertain to individuals, without contextual information about breeding colony, we use the term IFSF and avoid making interpretations that the require distinctions between IFSF and colony-level segregation of foraging site fidelity. To better contextualise this evidence for long-term IFSF focusing on areas off the coast of Vancouver Island, we can consider questions about both geographical (e.g., is it a feasible that this area could have had major hotspots in the past?) and behavioural (e.g., is there precedence for long-distance, long-term IFSF behaviour in other taxa?) contexts in closer detail.

First, although there are no major foraging hotspots known in the region today, the historical and archaeological records indicate that there may have been in the past. Historical observations from along the west coast of Vancouver Island and the Strait of Juan de Fuca are rare (though not surprisingly given the species' early disappearance), but they do indicate that short-tailed albatross' were both abundant (as suggested by their availability for sale in markets of the nearby city of Victoria c. 1852–1869 CE) and had a year-round presence in the region[58]. In combination with evidence for the dominance of short-tailed albatross among archaeological deposits at other sites in the area (e.g., Maplebank [DcRu-12] and Hesquiat Village [DiSo-1][59]), these observations provide strong support for the idea that short-tailed albatross recovered from the Yuquot site may have been attracted to the area in large numbers by local foraging hotspots that are no longer a focal point for the species.

Second, while long-term or multi-annual IFSF has not been observed among modern North Pacific albatrosses, there is some evidence to suggest not only that it could be feasible, but that its current absence could be linked with low population density. A recent tracking study has, for instance, provided the first demonstration of long-term IFSF behaviour (over a two-year period) among Campbell albatross (*Thalassarche impavida*) in New Zealand[25], demonstrating that repeated use of the same distant foraging areas is possible for long range migrants. Shorter-

term tracking studies with other seabird species have, moreover, shown that at smaller time scales IFSF may be an adaptive response to interspecific or conspecific competition (e.g.,[60,61]). With respect to short-tailed albatross and foraging behaviour in the study region in particular, though multi-year tracking studies are still relatively few in number, there has been some evidence for juvenile (though not adult) foraging mobility that is focused on a patch of shelf-break habitat off the western coast of Vancouver Island, but use of this has mainly been observed on a limited seasonal basis[12]. In this context, the absence of long-term IFSF today may reflect low competition among short-tailed albatross.

Unfortunately, biological ages of specimens could not be assessed in this study (due to a lack of regression formulae for relevant skeletal elements) and we therefore cannot directly assess the extent to which our findings apply to adult vs. juvenile short-tailed albatross behaviour. For this reason it remains possible that the long-term IFSF we have observed may have had a strong degree of life history variation. For instance, it could be that long-term IFSF was a strategy that was primarily used by younger individuals that later moved or diversified their geographical foraging focus in adulthood. Recent tracking studies and sighting records, for instance, suggest that while sometimes used by juveniles, the region is rarely used by foraging adults today[7,12]. In contrast, estimated ages profile for short-tailed albatross in the region based on historical observations indicate equal parts juveniles and adults (based on plumage), though the authors also note that identification issues may have led to under-reporting juveniles[58]. These data at least go to show that adult short-tailed albatross were historically present and were hunted by Indigenous peoples in the nearby Strait of Juan de Fuca (Fig. 1). Our data from historical taxidermy mounts, which include both adult and juvenile individuals, can also speak to this question. Though the timeframes integrated by bone and feather differ significantly (and are therefore not directly comparable[34]), isotopic compositions from both these individuals appear to cluster with the archaeological data from bone collagen (Supplementary Fig. 3), suggesting that the behavioural patterns observed among short-tailed albatross harvested at the Yuquot site were feasible for both older and younger individuals.

**Conservation implications**. Regardless of the mechanisms underlying foraging behaviour observed today, our data suggest that, at a minimum, short-tailed albatross are both capable of long-term IFSF and that this behaviour could be focused on relatively small areas at distant edges of their foraging range. Assuming that these behaviours were in some way guided by density-dependent competition[60,62,63], it also seems likely that long-term IFSF could become common among some short-tailed albatross once again when populations approach carrying capacity. This finding may, in turn, have implications for conservation. First, if a high degree of long-term IFSF were to re-emerge among a subset of the short-tailed albatross population (be they unconnected individuals or individuals from the same colony), it would mean that some birds could become more vulnerable to hazards in localised areas. Second, our data suggest that areas selected as hotspots can occur in places that do not currently play a major role in the short-tailed albatross' foraging range. In either case (i.e., development of new hotspots or long-term IFSF), our data highlights a need for further monitoring of modern short-tailed albatross foraging behavioural changes that could increase exposure to hazards as well as continuing to develop mitigation strategies. Coupled with genetic analyses (to assess the extent to which evolving behaviour could be hereditary), further multi-year

tracking studies that follow birds over a longer period of their lifespan (from juveniles into adulthood), especially as populations numbers grow, would provide a means of monitoring the presence, driving forces, and locational sensitivities associated with this behaviour.

**Implications for understanding broader seabird behaviour**. Our isotopic evidence also has broader implications for our understanding of seabird behaviour and conservation in general. Owing to technological and other difficulties in conducting multi-year tracking studies with the same individuals, evidence for long-term IFSF has only recently been discovered among seabirds living today[23], and little is known about the extent of this behaviour across different taxa or in relation to key spatiotemporal variables. Observation of IFSF spanning multiple breeding seasons is significant as it implies that foraging site specialisation is linked either with genetics (i.e., is innate) or with learned site familiarity, which, in either case could allow individuals to develop knowledge specific to particular areas that offers a competitive advantage[23,64,65]. Our data provide the first record showing the consistent presence of long-term IFSF in a species, not just on a scale of a few years or even decades, but for millennia. While it is difficult to assess the extent to which this behaviour was driven by learned and innate mechanisms, or whether it occurred among individuals from one or more colonies, the consistency and deep antiquity of this behaviour suggests that innate drivers are possible and that the behaviour could have become prevalent among individuals from specific colonies (i.e., developed into a colony-level affinity of foraging site fidelity). Ongoing genetic analyses of the same specimens may provide new insights. Among our samples, the absence of substantial isotopic variation that has been observed in other regions (Fig. 2a, c–f) further implies the presence of competitive exclusion of more widely foraging birds (i.e., those following a win-stay, loose-shift strategy[66]) and, in turn, provides support for the idea that development of long-term IFSF was density-dependent in this region of the short-tailed albatrosses foraging range.

Mechanisms (i.e., genetic vs. learned) and driving forces (e.g., competition) aside, this observation of long-term IFSF is further remarkable in that occurred over enormous distances (between breeding and foraging sites that are over 7000 km apart) and was apparently stable over hundreds of generations of short-tailed albatross. These extremes may suggest that this foraging strategy could be a more fundamental adaptation not only among this species, but for many other long-distance migrants. As highlighted in the first instance of its discovery by Wakefield and colleagues[23], the existence of long-term IFSF has important and potentially wide-ranging implications including for future study design (e.g., need for longer-term satellite tacking[23]), metrics for environmental change (e.g., reliability of seabirds as indicators for marine environment monitoring[67]), and our understanding of links between environment and age in determining behaviour (e.g., the lifetime consequences for conditions experienced in early development[68]).

**Sustainability of indigenous seabird harvesting practices**. Lastly, but perhaps most importantly, our findings also speak to the deep antiquity of sustainable Indigenous marine resource harvesting. Archaeological and ethnographic evidence shows that Indigenous communities along the Northwest Coast of North America have utilised seabirds, including short-tailed albatross, for many millennia (e.g.,[59]), but these data have a limited capacity for assessing how long-term exploitation may have influenced bird behaviour, which could be a key indicator

for understanding harvest sustainability. Short-tailed albatross was the single most abundant avian taxa across archaeological deposits at the Yuquot site (representing 1866 of 5966, or 31%, of the bird specimens) and was, therefore, an important resource for community members[69]. Unfortunately, by the time ethnographic recordings of bird hunting were made (1870–1900 CE) in the local study region[70], short-tailed albatross presence in the area was likely already heavily impacted by hunting at their breeding colonies[69]. It is therefore not surprising that, in contrast to other economically important bird taxa, information about the capture techniques and even the presence of short-tailed albatross in the region were not recorded in the course of early ethnographical work[70]. It is possible that, at points during this 4250-year record of harvesting seabirds at the Yuquot site, the population of short-tailed albatross that specialised in foraging in the region could have been depleted. Moreover, birds with high affinity for long-term IFSF focusing on local hotspots might be particularly vulnerable to over exploitation, a scenario that would reduce competition in the area and provide new opportunities from 'geographical generalist' short-tailed albatross (i.e., those that did not use a forging strategy focused on a specific area) to utilise the region. However, in this scenario, we would expect the occurrence of over exploitation to lead to greater isotopic variation, which would be associated with inclusion of more 'geographical generalist' birds being included in the harvest. In that context, our results suggest that, regardless of what hunting approaches were used to collect short-tailed albatross by members of the Yuquot community, Indigenous harvesting likely did not have a significant impact on bird numbers and therefore was sustainably practiced. Our data is consistent with previous archaeological and ethnographic studies that have suggested long-term Indigenous stewardship of marine resources on the Northwest Coast of North America (e.g.,[71]).

## Methods

**Sample description**. Archaeological short-tailed albatross bone samples ($n = 95$) are from the Yuquot site, a Mowachaht Nuu-chah-nulth village located on Nootka Island off the west coast of Vancouver Island. Excavations of the site uncovered well-stratified archaeological deposits rich in short-tailed albatross remains, which are divided into four radiocarbon-dated stratigraphic zones[69,72]. From earliest to latest these are zones 1 (pre-2300 to 1000 BCE), 2 (1000 BCE to 800 CE), 3 (800 to 1789 CE), and 4 (1789 to 1966 CE)[72]. Taxonomic identification of archaeological specimens utilised faunal reference collections from four repositories (University of Florida in Gainesville, Florida, USA; the Royal Ontario Museum in Toronto, Ontario [ON], Canada; the National Museum of Natural Sciences in Ottawa, ON, Canada; and the University of California in Berkeley, California, USA) and are considered to be robust[59,69]. Archaeological materials from Yuquot are currently curated at the Canadian Museum of History in Gatineau (Quebec, Canada). Permission to analyse the archaeological specimens was provided by the Mowachaht/Muchalaht First Nation in Tsaxana (BC, Canada) and Parks Canada in Ottawa (ON, Canada). Feather samples were taken from taxidermied adult and juvenile short-tailed albatrosses ($n = 2$) curated at the Royal British Columbia Museum in Victoria (BC, Canada) that were collected in the Strait of Juan de Fuca in 1889 CE. Detailed descriptions of these taxidermied specimens are provided by Carter and Sealy[58]. Permission to sample these specimens was provided by the Royal British Columbia Museum. Provenience information for the archaeological and historical samples is provided in Supplementary Data 1.

**Sample preparation**. Bone samples were selected based on minimum number of individual counts per archaeological context to minimise the possibility of duplicating data from the same individual. Bone collagen extractions followed well established protocols[73] in the Archaeological Chemistry Laboratory (ACL) at the University of British Columbia in Vancouver (BC, Canada; e.g.,[74]). Samples were cut into small $2 \times 2$ mm cubes, demineralised in 0.5 M hydrochloric acid (HCl) over several days, and then rinsed to neutrality in Type 1 water (resistivity of >18 $M\Omega$ cm). Demineralised samples were then soaked in 0.1 M sodium hydroxide in an ultrasonic bath (solution refreshed every 15 min until solution remained clear) to remove base-soluble contaminants and then again rinsed to neutrality in Type 1 water. Samples were then refluxed in 0.01 M HCl at 65 °C for 36 h and centrifuged to separate out insoluble residue. Collagen was pipetted to a fresh tube, frozen, and

lyophilised. Contour feather tip samples were removed from mounts using scissors. Feather samples were cleaned prior to analyses with a 24 h soak in 2:1 chloroform methanol[75] before being air dried and then rinsed three times in Type 1 water in an ultrasonic bath.

**Statistics and reproducibility**. Isotopic compositions were measured on $CO_2$ and $N_2$ gases produced by combustions of 0.5 mg subsamples of collagen or feather in a Vario MICRO Cube elemental analyzer coupled to an Isoprime isotope ratio mass spectrometer (Elementar, Hanover, Germany) at the ACL. Replicate analyses were performed on 19% of samples. Calibrations of isotopic compositions were made relative to AIR for nitrogen and VPDB for carbon using USGS40 and USGS41a[76,77]. Monitoring of accuracy and precision was accomplished using five internal standards. Known (for calibration standards) or long-term observed averages (for check standards) of all isotopic reference materials are shown in Supplementary Table 4. Averages and standard deviations for calibration standards (Supplementary Table 5), check standards (Supplementary Table 6), and sample replicates (Supplementary Table 7) for all analytical sessions are also available in the supplementary materials. Following Szpak and colleagues[78], for $\delta^{13}C$ and $\delta^{15}N$: systematic errors [$u_{(bias)}$] were ±0.09‰ and ±0.15‰; random errors [$uR_{(w)}$] were ±0.06‰ and ±0.13‰; and standard uncertainty was ±0.11‰ and ±0.19‰. Collagen QC was established using carbon (>13.8%) and nitrogen (>4.0%) elemental concentrations[55] as well as liberal $C:N$ ratio criteria[56]. We also looked for a relationship between collagen $C:N$ and $\delta^{13}C$, which may be detectable when contamination-altered isotopic compositions are present among larger intra-specific datasets[56].

Statistical tests were performed using PAST version 3.22[79]. Potential for a relationship between $C:N$ and $\delta^{13}C$ was assessed using a Pearson's correlation coefficient. Presence of chronological patterns among samples from temporal zones 1 through 4 was assessed by comparing group means. Normality of distribution and homogeneity of variance were assessed using Shapiro–Wilk and Levene's tests, respectively. For normally distributed groups where variances were not determined to be equal, a Welch's $F$-test was used to compare group means. In cases where one or more groups weren't normally distributed, comparisons were made with a Bonferroni-corrected Mann–Whitney $U$-test. A $p$-value of 0.05 or less was considered significant. Using the SIBER package[54] in R version 3.6.0[80] through RStudio version 1.2.1335[81], standard bivariate ellipse area (SEA), standard ellipses area with Bayesian estimate ($SEA_B$), standard ellipse area corrected for sample size ($SEA_C$), as well as total area (TA; also known as convex hull area) were used to quantify isotopic variation among different temporal and geographical groups. The default parameters were used to calculate $SEA_B$ (see SIBER package description for more information).

Our statistical analyses are oriented towards distinguishing between degrees of isotopic variation and do not compare inter-site group means because, due to potential overlap between regional isotopic baselines, these comparison would not necessarily be suitable for testing our hypotheses. For instance, although demonstrating a statistical difference between mean isotopic compositions among sites could be useful, failure to demonstrate a statistical difference between groups does not necessarily mean that they did not primarily forage in different regions. In contrast, comparing inter-site group variation does provide a way of demonstrating larger-scale patterns in foraging behaviour.

**Reporting summary**. Further information on research design is available in the Nature Research Reporting Summary linked to this article.

## Data availability

All data used in this study are provided in the Supplementary Data and Supplementary Materials files associated with the published article.

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

## Acknowledgements

Sampling permissions and assistance were provided by Gavin Hanke and Lesley Kennes at the Royal British Columbia Museum and Charles Dagneau, Janet Stoddard, Mike Steinhauer, and Nancy McCarthy at Parks Canada. Ken Morgan at Environment and Climate Change Canada (Sidney, BC, Canada) provided access to unpublished literature. Sampling and technical assistance was provided by Dylan Hillis at the University of Victoria (Victoria, BC, Canada) and Jess Metcalfe at UBC. Funding for analyses was provided by the UBC's Department of Anthropology.

## Author contributions

Conceptualization: E.G.; Methodology: E.G.; Resources: E.G. and M.J.; Investigation: E.G., M.J., C.C., and T.C.A.R.; Visualization: E.G., C.C., and T.C.A.R.; Project administration: E.G.; Writing (original draft): E.G.; Writing (review and editing): E.G., T.C.A.R., M.J., and C.C.

## Competing interests

The authors declare no competing interests.
