## [Peer Review File · Communications Biology]

Reviewers' comments:

Reviewer #1 (Remarks to the Author):

The authors use isotopic data from a 4,000 year series of short-tailed albatross archaeological remains from western Vancouver Island to provide evidence of long term foraging site fidelity. The authors put these data into a larger framework of other short-tailed albatross isotopic studies from other sites including Oregon, California, Japan, and Russia. The Vancouver site provides evidence of foraging site fidelity based on the limited isotopic variation through time.

First and foremost, thank you for writing a well written manuscript. You made my job as a reviewer easier. Studies across long time frames prior to European arrival are critical to understanding human impacts on diversity and distributions across the Holocene. These studies (as you acknowledge) as also important for conservation initiatives.

I cannot claim to be an expert in isotopic data, so I must leave any comments regarding the analyses to other reviewers. I will say that based on looking at the raw data in the supplement and Table 2 the range of values and the standard deviation from each locality seem very similar and overlap in some cases. Are these sites significantly different from each other in terms the isotopic values? I am convinced that they're different based on the simulation but would like to know if a p value supports those differences.

Overall, it would be helpful to see the radiocarbon date error bars associated with each sample in the supplementary data. Alternatively, if it is already represented in the table within the defined time bin e.g., "2300-1000BCE" then please explicitly add that information to the main text or to the caption.

Discussion points

I have thought hard about other mechanisms that could drive the isotopic pattern in your study. One thought concerns sea currents and upwellings. Since upwellings are nutrient rich mixed waters (with presumably mixed isotopic signatures) I would suspect that along the California and Oregon coast, sea shelf foragers would have well mixed isotopic signatures regardless of their foraging site fidelities. The Washington-Vancouver border looks like a boundary for the California and North Pacific Current. Can you speak to how the currents and upwellings across the past 4,000 might be driving your patterns? Is there also an upwelling off Vancouver? Perhaps historically all individuals from sites sampled had limited foraging behavior and upwellings are driving the isotopic variation?

Also, are there any ideas about the impact of overharvested, unhealthy fisheries on foraging site fidelity regardless of population size? Based on current ocean health, I would suspect even with healthy population sizes modern sea birds are going to have to go far and wide to find reliable food sources.

If the Yuquot population was behaviorally different from other populations of short-tailed shearwaters I am looking forward to finding out if it was genetically distinct population based on ancient DNA. (This is really a comment and doesn't require a response.)

Minor points:

Line 105 - should be severe

Reviewer #2 (Remarks to the Author):

This submission uses the isotopic composition of bone collagen to infer foraging site fidelity in a large seabird. It relies on an impressive collection of material from archaeological site that spans

over 4000 years of activity. Work of this nature has been conducted on this species before and indeed many of these samples feature in this manuscript. As such the novelty here largely hinges on a single, but rather interesting result: the isotopic niche width at a single site is smaller (as measured using the package SIBER), than all other populations measured. If this were a contemporary study, then it would be of limited interest, but the timespan does make it worthy of consideration in Communications Biology. However, there is much more ambiguity around the key result than the authors seem to suggest and as such I am unconvinced about the contribution this work might make. There are also a number of minor issues that need attention.

My first concern is that the individual foraging site fidelity (IFSF) is not the same as colony-level segregation (the latter is what the observed patterns may describe given the caveats outlined below). Of course, the actual definitions of these are rather vague and dependant to some degree on the scale at which they are observed, but the phenomena driving them can be different (see any of the Wakefield et al studies you cite) and as such how they are interpreted also differs. This means that in various places the MS, population-level phenomena seem to be interpreted in an individual manner (I accept this may not be the intention) and this needs to be addressed.

My second concern is around the use of collagen to ascribe IFSF. The authors do a nice job of explain why collagen integrates very long periods (years) of foraging and thus is an appropriate tissue for such a study, but my feeling is the opposite. Contemporary work on seabirds, including the study species featured here, highlights how these birds range widely once freed up from the constraints of central place foraging. We know from this work that IFSF can occur over short temporal scales and within seasons. However, there is little evidence among wide ranging seabirds that IFSF is maintained across seasons (ie from summer to winter). Indeed short-tailed albatross work shows that these animals have huge ranges and move 1000s of kilometers during their annual cycles. The problem with collagen is that it 'averages out' these movements combining isotopic information from all seasons and as such signals of foraging site fidelity (which will almost certainly vary from one part of the annual cycle to the next as outlined above) get quenched. To this end, there is a huge literature that outlines the idea behind temporal period of isotopic integration and the temporal period of the ecology that is of interest which is largely uncited.

My final concern is that the data are much more equivocal than the authors suggest. This is in part down to the use of collagen and in part down to the difficulty of dealing with all the confounding variables that are likely present in such a disparate data set (this last part is not a criticism, I understand the limitations you have).

Likewise, there needs to be a recognition that isotopic similarity does not equal ecological similarity. Ocean isoscapes vary at small and large scales, so different ecological responses can generate very similar isotopic patterns. As such, a smaller isotopic niche measured in this way does not necessarily indicate smaller and more narrowly defined foraging areas than birds collected at other locations. It just means they are more like one another; this tells us nothing about IFSF, these birds could have huge ranges, but the collagen signal averages out the variation. If all birds ranged randomly across isotopic space with no site fidelity the population data would suggest a smaller isotopic niche than a population of birds with very small home ranges and high IFSF that did not overlap.

Neither do these data unequivocally suggest that foraging habitats were stable over 4,250 years, only that they foraged in isotopically similar sites, which is not necessarily the same. Of course we would expect them to forage in the same types of places, but this is not IFSF. The inference as I read it (and I am sorry if this is doing you an injustice) is that birds are maintain long-term colony-level foraging site fidelity, yet we know that these regions have not been climatically stable over the past 5,000 years and so this would be highly unlikely given the difference we observed from one year to the next in the same populations tracked from the same colonies. Arguing that the one would expect to see similar patterns in the nearest sites California and Oregon if it were not fidelity is not that helpful as we know nothing of the expected isotopic distributions that these birds were foraging in (contemporary data sets can show marked differences between colonies, despite potentially overlapping home ranges depending on colony location... you even cite one of these studies). These also represent distinct time periods within the Yuquot data set as well so expectations are unclear in that respect as well. Likewise it is easy to imagine how mechanisms of harvesting might change among the regions e.g young on nests vs adults at sea which also makes interpreting these local difference difficult.

Some minor issues:

I felt the MS lacks a bit of focus in places, it is set up in a conservation context, but not exactly

clear until the very end how this new info informs management etc, I think it could be tighter and a bit more clearly structured in this sense.

There are some sweeping statements, particularly in the intro that need attention e.g. the first, sentence of abstract does not really make sense, it is competition (which of course is a product of population size and resource availability) that is thought to be a major driver of IFSF, along with social facilitation and this it is not well understood. Likewise, the statement on increasing population size and hotspot expansion is just one potential response, there are others.

Isotopic niche is a well described concept and has been for around 15 years, it does not need inverted commas, just cite a source.

You need to be careful of contradictory arguments. For example, one of the rationales for the MS is that contemporary understanding of the ecology of this species is biased by the fact that it is a recovering population, but then you use contemporary foraging information to argue that the drivers of variation in stable isotope ratios is about foraging and not space

There are a lot of types e.g. sever v severe line 105

Reviewer #3 (Remarks to the Author):

This paper reports a detailed stable isotope analysis of bone collagen from archaeological specimens of the Short-tailed albatross recovered from the Yuquot Site on Vancouver Island, B.C. The bone samples range in age over the past 4250 years. Bulk carbon and nitrogen was completed on 95 bones from four distinct time zones within this period to determine if there had been temporal variation in foraging zones over this period. The authors place this study into the context of the modern collapse of this species' populations in the north Pacific. In addition, they compare their results with four other regional studies on this species to show that the Yuquot sample had a distinct foraging area, possibly related to fishing 'hotspots' that once existed near Vancouver Island.

The study is well designed and the paper is well written and thorough in their analyses and discussion. I had only a few suggestions that should be addressed. First, the bone samples analyzed indicate that used a variety of skeletal elements to extract bone collagen, including the tarsometatarsus, carpometacarpus, coracoid, and humerus. No analysis of isotopic results by different skeletal elements was provided and I think it would be helpful to do so, to show that the isotopic signals are the same regardless of element used in the analysis. Since some of these elements do vary in their development and degree of use in life, it would be important to include this analysis here. I also note that the study on the bones from Japan used the same bone (carpometacarpus) for each measurement, probably to avoid any inter-skeletal variation in their results.

Second, the authors could have included analysis of sulfur isotopes to further differentiate foraging areas by time period. Sulfur is becoming increasingly useful in this regard and studies have shown that carbon and nitrogen may not vary significantly by time zone or region, but when adding sulfur another dimension in foraging history is revealed. It may be that there was not enough bone collagen from these bird bones to run that analysis along with carbon and nitrogen so I only mention this for future consideration. In addition, regarding the analysis by time zones, did the authors consider the Suess effect for the modern time zone 4? If this effect was negligible in the north Pacific, this should be mentioned in the methods as it could influence their comparisons of that time zone with the earlier periods.

Third, the discussion is rather long (7.5 pages) with no subheadings to help break up the different topics outlined here. It would be helpful to organize this section a little better using subheadings and avoid some of the repetition within it. I found myself getting a little lost in which topic was most important in their interpretations.

The figures and tables are useful and important for the study. Note, though, that there is a typo in Fig. 1: Jaun de Fuca should be Juan de Fuca. The caption for Table 1 should mention that there is no statistical differences in carbon and nitrogen among the time periods. The caption for Table 2

should indicate which means are statistically different (I think the low mean carbon value from Japan is significantly different from the sample from Oregon, and nitrogen from the sample from California?).

Reviewer #1 (Remarks to the Author):

The authors use isotopic data from a 4,000 year series of short-tailed albatross archaeological remains from western Vancouver Island to provide evidence of long term foraging site fidelity. The authors put these data into a larger framework of other short-tailed albatross isotopic studies from other sites including Oregon, California, Japan, and Russia. The Vancouver site provides evidence of foraging site fidelity based on the limited isotopic variation through time.

First and foremost, thank you for writing a well written manuscript. You made my job as a reviewer easier. Studies across long time frames prior to European arrival are critical to understanding human impacts on diversity and distributions across the Holocene. These studies (as you acknowledge) as also important for conservation initiatives.

I cannot claim to be an expert in isotopic data, so I must leave any comments regarding the analyses to other reviewers.

1. I will say that based on looking at the raw data in the supplement and Table 2 the range of values and the standard deviation from each locality seem very similar and overlap in some cases. Are these sites significantly different from each other in terms the isotopic values? I am convinced that they're different based on the simulation but would like to know if a p value supports those differences.

This is a keen observation but can be resolved with a brief clarification. The interpretations we've made hinge primarily on the differences in isotopic variation (quantified through standard ellipse areas [SEA]) and do not directly involve statistical inter-site comparisons of between group means. The reason we have not included these comparisons is that the meaning behind statistical differences (or lack thereof) in inter-site group means is vague and difficult to interpret for such a highly migratory taxon. It is entirely possible, for instance, that two groups with statistically indistinguishable mean isotopic compositions foraged in very differ places. In contrast, differences in group variation can (as we have done here by quantifying SEAs to characterize isotopic feeding niches) more easily be used to distinguish between foraging location variability.

2. Overall, it would be helpful to see the radiocarbon date error bars associated with each sample in the supplementary data. Alternatively, if it is already represented in the table within the defined time bin e.g., "2300-1000BCE" then please explicitly add that information to the main text or to the caption.

As outlined in Sample Description section, the chronology for these samples is based on well-dated site stratigraphy. The site was highly stratified, meaning that radiocarbon dates on other materials (it is best to avoid dating samples form marine taxa due to varying reservoir effects) were able to peg each of our samples to a generalized temporal zone based on the deposit in which it was found. To make this clearer we have modified the caption for Table 1 to explicitly indicate this.

Discussion points

3. I have thought hard about other mechanisms that could drive the isotopic pattern in your study. One thought concerns sea currents and upwellings. Since upwellings are nutrient rich mixed waters (with presumably mixed isotopic signatures) I would suspect that along the California and Oregon coast, sea shelf foragers would have well mixed isotopic signatures regardless of their foraging site fidelities. The Washington-Vancouver border looks like a boundary for the California and North Pacific Current. Can you speak to how the currents and upwellings across the past 4,000 might be driving your patterns? Is there also an upwelling off Vancouver? Perhaps historically all individuals from sites sampled had limited foraging behavior and upwellings are driving the isotopic variation?

These are all interesting questions, which we can speak to here for clarification, but note that they are covered in our discussion of Vokhshoori, et al. (2019) study exploring baseline variation across the short-tailed albatross' range (see the Introduction section and Stable Isotope Theory and Interpretive Context section). Yes, foraging hot spots today, and very likely in the past as well, were centered around shelf break areas where upwelling of deeper, nutrient rich waters are brought to the surface. So there is strong relationship there. Yes, there areas of upwelling off the coast of Vancouver Island, not too far from the Yuquot site and it is likely that it these are the areas that were a foraging focus for the short-tailed albatrosses caught by Yuquot residents. The same is true for birds caught at other comparative sites (e.g., Oregon and California). While upwelling can have a strong impact on isotopic compositions (our reference to Sigman, et al. 2009 is a good place to start for looking at this) there is a wider suit of variables (namely pertaining N and C cycling parameters and sources) that govern baseline isotopic compositions. In that context, we (and others who have workd on the comparative data: Eda, et al. 2012; Vokhshoori, et al. 2019) don't anticipate that any upwelling specific variables would be governing the patterns that we have observed.

4. Also, are there any ideas about the impact of overharvested, unhealthy fisheries on foraging site fidelity regardless of population size? Based on current ocean health, I would suspect even with healthy population sizes modern sea birds are going to have to go far and wide to find reliable food sources.

Yes there is literature on this, but with respect to short-tailed albatrosses specifically, the picture is complicated by the fact that the species is still recovering from a near-extinction event and is still not nearly as abundant in part of its range as it once was. This means that considering impacts form over harvesting of the ocean food webs upon which this species depends is not possible at present.

5. If the Yuquot population was behaviorally different from other populations of short-tailed shearwaters I am looking forward to finding out if it was genetically distinct population based on ancient DNA. (This is really a comment and doesn't require a response.)

We agree and look forward to publishing the results of our ongoing genetic analyses of the same samples.

6. Minor points: Line 105 - should be severe

This change has been made.

Reviewer #2 (Remarks to the Author):

This submission uses the isotopic composition of bone collagen to infer foraging site fidelity in a large seabird. It relies on an Impressive collection of material from archaeological site that spans over 4000 years of activity. Work of this nature has been conducted on this species before and indeed many of these samples feature in this manuscript. As such the novelty here largely hinges on a single, but rather interesting result: the isotopic niche width at a single site is smaller (as measured using the package SIBER), than all other populations measured. If this were a contemporary study, then it would be of limited interest, but the timespan does make it worthy of consideration in Communications Biology. However, there is much more ambiguity around the key result than the authors seem to suggest and as such I am unconvinced about the contribution this work might make. There are also a number of minor issues that need attention.

1. My first concern is that the individual foraging site fidelity (IFSF) is not the same as colony-level segregation (the latter is what the observed patterns may describe given the caveats outlined below). Of course, the actual definitions of these are rather vague and dependant to some degree on the scale at which they are observed, but the phenomena driving them can be different (see any of the Wakefield et al studies you cite) and as such how they are interpreted also differs. This means that in various places the MS, population-level phenomena seem to be interpreted in an individual manner (I accept this may not be the intention) and this needs to be addressed.

We appreciate that the distinction between individual- and colony-level foraging site fidelity (FSF) is complex. In many ways, given what is known about how short-tailed albatross behave today, interpreting our results as evidence for colony-level segregation would actually be a lot more interesting than IFSF from a conservation perspective. As we outline below, while both are possible, our evidence is derived from individuals and, though we examine the data as a group, the isotopic compositions ultimately pertain to individuals only. In other words, without knowing the relationships between colony origins of each bird in our study, it would be a big leap to claim that we can assign the behaviours we've uncovered to the colony-level. However, in the context of this study (i.e., dealing with a long time frame, extreme distances between colonies and our study site, and focusing of behaviour of birds from colonies that may no longer exist), the differences between these two levels of behaviour do not significantly impact our findings or their implications. There are a couple of reasons for this.

First, FSF at a colony level is necessarily the result, at least initially, of individual behaviours and will, at some point, have to have been derived from select individuals' foraging patterns that were then passed on to others. So whether the phenomena we've observed here are reflecting FSF by individuals from different/multiple colonies or colony-level FSF (or just a portion of one colony), it still demonstrates that short-tailed albatross (and possibly other sea birds) possess the ability for IFSF (i.e., colony-level FSF is rooted in IFSF at some point).

Second, as we argue, comparing our data with samples from sites in other regions, strongly suggests that the implications (e.g., likely presence of competitive exclusion or competitor avoidance) would still be an important factor in driving the

observed FSF regardless of whether it was something practiced by individuals from one or multiple colonies. The fact that much more isotopic variation is found at sites in other regions suggests that individuals with a wide range of foraging behaviours were being collected locally at these sites. In contrast, the fact that only birds with a narrow isotopic niche were collected at the Yuquot site suggests that the birds available locally in the area were not feeding in the kind of wide ranging environments that were seen from other areas. Because there could be no way for residents of the Yuquot site to distinguish between birds with different migratory behaviours, this is very strong evidence that short-tailed albatross with wider foraging tendencies were not available for collection in the area (i.e., non-regionally-specialised birds were excluded from or avoided the area where Yuquot residents hunted due to strong existing competition).

We would also highlight that we have outlined that this behaviour could be genetically driven, which, in the context of how short-tailed albatross breed today (low levels of inter-colony breeding), at least implies that this foraging behaviour could be practiced by individuals from a subset of the population (allowing for the possibility of colony- or sub-colony-level segregation). As a final potentially relevant point, we note that there is no credible evidence for the past existence of a breeding colony in the eastern Pacific. This confirms the very long-distance nature of this FSF, whether inter- or intra-colony, and that it is not the result of a now-extinct colony of birds that simply fed differently/more locally than is observed today.

The above notes (which we believe are clearly argued in the manuscript) notwithstanding, we have added new text to the Discussion section (in multiple areas but mainly see Lines 281-289) outlining the possibility that the patterns we see could result from behaviour of birds at one colony.

2. My second concern is around the use of collagen to ascribe IFSF. The authors do a nice job of explain why collagen integrates very long periods (years) of foraging and thus is an appropriate tissue for such a study, but my feeling is the opposite. Contemporary work on seabirds, including the study species featured here, highlights how these birds range widely once freed up from the constraints of central place foraging. We know from this work that IFSF can occur over short temporal scales and within seasons. However, there is little evidence among wide ranging seabirds that IFSF is maintained across seasons (ie from summer to winter). Indeed short-tailed albatross work shows that these animals have huge ranges and move 1000s of kilometers during their annual cycles. The problem with collagen is that it ‘averages out’ these movements combining isotopic information from all seasons and as such signals of foraging site fidelity (which will almost certainly vary from one part of the annual cycle to the next as outlined above) get quenched. To this end, there is a huge literature that outlines the idea behind temporal period of isotopic integration and the temporal period of the ecology that is of interest which is largely uncited.

We appreciate the Reviewer 2’s concern here and like to offer a comment on how we cite the literature. We are intimately aware of the literature on how different tissues integrate isotopic compositions overtime (in fact, EG and CC have published a number of papers on this topic). Unfortunately, the citation limit for our initial

submission to Nature Communications (which was directly transferred to Communications Biology) meant that we could only cite the core papers. While there are a number of bone collagen specific papers we could choose from, we chose to cite Hobson and Clark's seminal (1992) work as it is not only one of the first papers to demonstrate how bone collagen turnover differs from other tissue, but is also bird specific. We have now added a new reference to Hyland, et al. (2021), which also includes bird bone collagen data, to provide an example of the latest relevant reference.

We would also like to take this opportunity to further contextualize the parts of the manuscript that describe how we can use bone collagen isotopic compositions to ascribe variation in foraging fidelity. Bone collagen is a tissue that has been the focus of thousands of archaeological studies with 100,000+ analyses to date, but has been less prevalent in the ecological literature (although ecological use of bone/tooth collagen isotopic data does continue to grow). In this context, it is possible that researchers in other disciplines may be less aware of the degree of confidence that we can have in our interpretations of foraging temporality from bone collagen data. Although controlled studies on the finer details of the precise time periods reflected by bone collagen isotopic compositions have been few and far between (they are particularly costly/challenging to perform due to the long time frames involved), it is well established and widely accepted that bone collagen isotopic compositions reflect diet over at least several years (and in some cases multiple decades – as shown by our reference to Hedges, et al. 2007). For this reason, there is a significant averaging effect such that when isotopic differences are observed between bone collagen samples (or in this case, groups of individuals) we can say with certainty that this reflects meaningful long-term behaviour (diet/mobility) differences.

We agree with Reviewer 2 that if past short-tailed albatrosses were all evenly circulating and feeding across their entire range, we could expect to see relatively little isotopic variation between bone collagen from different individuals and sites. However, as Vokhshoori, et al. (2019), along with Eda, et al. (2012), have already demonstrated using archaeological bone collagen isotopic data from sites around the North Pacific, this is not the case. The reality of the situation is far more complex. In particular, Vokhshoori's bone collagen isotopic data from other sites demonstrates that, despite the tremendous distances and variation in migratory behaviour exhibited by this species, the typical isotopic variation associated with these birds is relatively wide ranging. In other words, in contrast to the reviewer's assumption (i.e., that the averaging effect of slow turn over rates in bone collagen will 'quench' isotopic variation associated with a lifetime of high mobility, resulting in a narrower range of variation at the population or sub population level) these data demonstrate that the ranging behaviour of short-tailed albatross varies between individuals such that large-scale, lifetime average differences are common between individuals despite the time-averaging effects of the slow turnover rates on bone collagen isotopic compositions. In this context, the pronounced differences we have seen in the isotopic variation between short-tailed albatross from the Yuquot site and those from other regions demonstrates clearly that the birds collected in these different areas had different behavioural patterns at the lifetime scale.

In that context, and building on Vokhshoori's isoscape research (showing consistent inter-regional ocean baseline differences across the short tailed albatross' range - more below), collagen isotopic compositions provide a powerful tool for differentiating between FSF of individual birds represented in the archaeological record. Though it is technically still possible that the patterns we have observed could result from Yuquot birds having only come from a sub group that consistently/evenly used a wider foraging area (creating a less variable group of isotopic compositions), the odds of this pattern being something that could naturally occur in our data are extremely unlikely. Our sample is larger and more temporally varied than that from any of the other regions examined by Vokhshoori and Eda's papers, meaning that we should be more likely to pick up on variation. Despite this much larger sample, as shown by our bootstrapping analyses of the entire dataset of published short-tailed albatross bone collagen isotopic compositions, across 1000 iterations for a sub sample comparable to our sample size we could find only one example of a similarly low incidence of variation for $\delta^{15}\text{N}$ and none for $\delta^{13}\text{C}$. Put another way, it is extremely unlikely that our sample simply reflects a unique circumstance where Yuquot residents only harvested short-tailed albatross with a specific, highly (isotopically) homogenizing, and wide-scale foraging behaviour. We believe that we have clearly outlined this in the manuscript.

3. My final concern is that the data are much more equivocal than the authors suggest. This is in part down to the use of collagen and in part down to the difficulty of dealing with all the confounding variables that are likely present in such a disparate data set (this last part is not a criticism, I understand the limitations you have).

We are uncertain of what Reviewer 2 is specifically pointing to here. On the one hand, this pair of sentences appears to have been entered as a separate paragraph, suggesting that it might require a separate response. On the other hand, the following sentences (i.e. listed here as Comment 4) seem as though they are linked. Assuming this is a standalone comment, we'd like to highlight that positive, unequivocal evidence for any phenomenon in the archaeological record is extremely rare and such claims should (rightly) always be met with scrutiny. We have been careful to interpret our results in a conservative way. As outlined in our response to Reviewer 2's Comment 2, our data provides very strong evidence that the short-tailed albatrosses harvested at the Yuquot site had foraging behaviours that differed markedly on average when compared to archaeological individuals collected in other regions. Our bootstrapping analyses demonstrate that it is extremely unlikely that these patterns result from random chance and, as mentioned, the fact that our data are from bone collagen drive home the point that these patterns represent persistent (lifetime) behavioural differences. Therefore, while we cannot be 100% certain of our interpretations (something that is impossible for research in many fields), we are confident that they reflect meaningful long-term patterns in short-tailed albatross FSF.

4. Likewise, there needs to be a recognition that isotopic similarity does not equal ecological similarity. Ocean isoscapes vary at small and large scales, so different

ecological responses can generate vary similar isotopic patterns. As such, a smaller isotopic niche measured in this way does not necessarily indicate smaller and more narrowly defined foraging areas than birds collected at other locations. It just means they are more like one another; this tells us nothing about IFSF, these birds could have huge ranges, but the collagen signal averages out the variation. If all birds ranged randomly across isotopic space with no site fidelity the population data would suggest a smaller isotopic niche than a population of birds with very small home ranges and high IFSF that did not overlap.

Here Reviewer 2 is, in part, (1) circling back to point made in Comment 2 (i.e., needing extra contextualization of how collagen isotopic compositions can reflect long-term foraging site fidelity) but also (2) suggests that we have not acknowledged/considered the complexities of processes (e.g., varying C and N cycles and source/inputs) that govern ocean isoscapes. We address these sequentially below.

First, as outlined in our response to Reviewer 2's Comment 2, data from Vokhshoori, et al. (2019) has already demonstrated that, while collagen has a long-term averaging effect on isotopic compositions, the foraging and mobility behaviour of short-tailed albatross still results in tremendous range of isotopic variation that reflects inter-individual differences in foraging mobility. In other words, while this may seem at odds with the ideas that these birds circulate wildly across the North Pacific and that bone collagen reflects a long-term average, Vokhshoori's findings go to show that there is enough foraging variation between individuals to result in very pronounced inter-individual differences arising from baseline isotopic variation in prevailing feeding location.

Second, we would like to highlight that we are intimately aware of high degree of complexity that characterizes aquatic biogeochemical cycles (as cited in the manuscript EG has, for example, written a review [Guiry 2019] on the topic as it pertains to, arguably, more biogeochemically complex freshwater environments). We have outlined, with specific references (Laws, et al. 1995; Popp, et al. 1998; Rau, et al. 1989; Sigman, et al. 2009), some of the kinds of variables that are relevant. However, given the wide scope of the short-tailed albatross' foraging range, it would not be feasible to unpack each potential variable in detail. We also highlight the fact that, based on Vokhshoori's detailed analyses of both the regional isoscape and single amino acid isotopic compositions from archaeological and modern albatross bone collagen, we can anticipate that issues related to baseline variation will not be an issue for our interpretations. In particular, Vokhshoori's exploration of the relative importance of baseline (i.e., differing isoscape regions across the foraging range) vs. dietary (feeding on different kinds of food) variation clearly shows that the main drivers of differences in short-tailed albatross bone collagen isotopic compositions are foraging location associations (i.e., feeding in differing areas of the North Pacific).

We feel that the above responses to these two concerns are clearly outlined in the manuscript. As a final note, while we can appreciate that these critiques have arisen from legitimate concerns, having worked extensively with other higher level consumers in marine settings (fish, marine mammals, and birds), we are certain that

bone collagen isotopic compositions of migratory animals that move between different regions of heterogeneous isoscapes can often provide clear insights into foraging mobility. While it is true that not all situations (taxa, locations, mobility patterns) will lend themselves to this kind of investigation, the case presented for the short-tailed albatross in this manuscript is one clear example of where our approach will work well.

5.

- a. Neither do these data unequivocally suggest that foraging habitats were stable over 4,250 years, only that they foraged in isotopically similar sites, which is not necessarily the same. Of course we would expect them to forage in the same types of places, but this is not IFSF. The inference as I read it (and I am sorry if this is doing you an injustice) is that birds are maintain long-term colony-level foraging site fidelity, yet we know that these regions have not been climatically stable over the past 5,000 years and so this would be highly unlikely given the difference we observed from one year to the next in the same populations tracked from the same colonies. Arguing that the one would expect to see similar patterns in the nearest sites California and Oregon if it were not fidelity is not that helpful as we know nothing of the expected isotopic distributions that these birds were foraging in (contemporary data sets can show marked differences between colonies, despite potentially overlapping home ranges depending on colony location... you even cite one of these studies). These also represent distinct time periods within the Yuquot data set as well so expectations are unclear in that respect as well.

Here Reviewer 2 is questioning the extent to which temporal variation in isotopic baselines could be an alternative explanation for the patterns we've observed in short-tailed albatross bone collagen isotopic compositions. It is true that isotopic baselines can change in a given region over time and we have now added text that more clearly acknowledges this to the Stable Isotope Theory and Interpretive Context section (Lines 161-167). However, based on what we know about the processes that govern baseline variation across different ocean areas, we do not expect that the level of baseline variation in different regions of this isoscape would vary at a scale that could explain the variation observed across the existing archaeological and modern short-tailed albatross database. Indeed this is not a factor that either Vokhshoori, et al. (2019) or Eda, et al. (2012) (both in top tier specialist journals in this area) have devoted much text towards, as it is a far less parsimonious explanation (i.e., suggesting that isotopic variation on the order of several per mill for both $\delta^{13}\text{C}$ and $\delta^{15}\text{N}$ is best explained as evidence for baseline shifts across different regions of an entire ocean rather than simply reflecting variation in the mobility of individual birds). We would also highlight the manuscript's sections pointing out that if baseline variation were an explanation for why birds collected at other sites dating to the same time periods then we should also find it at the Yuquot site as well unless the Yuquot birds were behaving very differently. In other words, in suggesting that baseline variation could be an explanation for why some contemporaneous samples show more variation than in

our dataset, there is a clear implication that the birds collected in each area have systematically different foraging regions.

With respect to the final portion of this comment, about how contemporary datasets can show marked differences between colonies, we would refer the reviewer to our response to Comment 2 above. Furthermore, coupled genetic and isotopic data from archaeological albatross generated by Eda, et al. (2012) showed that individuals that were likely from multiple colonies (including those ancestral to the extant Izu and Senkaku colonies) were feeding across the same areas (i.e., had similar isotopic compositions and variation) for at least one region (western edge of the Pacific) of the short-tailed albatross' foraging range.

- b. Likewise it is easy to imagine how mechanisms of harvesting might change among the regions e.g young on nests vs adults at sea which also makes interpreting these local difference difficult.

Here the reviewer is questioning whether different harvesting strategies across comparative sites/regions could explain the variation we've observed. This is an excellent question and is one that we have tried to address already. Aside from bait luring at sea (followed by blunt- or sharp-force dispatching), we're not aware of other harvesting options that could be used to collect short-tailed albatrosses. Further, based on ethnographies from different regions along the NW coast of North America it is likely that this same strategy was used by all (note that all sites examined are not at breeding colonies). It is therefore unlikely that strategies for attracting and killing birds would have introduced selective biases into our sample. It is, however, conceivable that human decisions about which albatross to collect could have been selective and, if this were the case, it could theoretically be a mechanism for creating structured isotopic variation across our sample. In a nutshell, it is possible that preference for collecting juvenile vs. adult birds (or a greater likelihood that juveniles would take bait), could generate systematic isotopic patterning. The underlying reason for this would be that juveniles and adults could be foraging on different isotopic baselines and that the temporal averaging effects of bone collagen turn over between adults (longer) and juveniles (shorter) differ. In this context, juveniles would be expected to have more isotopic variation because the shorter period of bone collagen turn over means that they would be susceptible to showing aberrant isotopic patterns associated with any idiosyncratic feeding behaviours in a more pronounced way. However, we have argued this this is unlikely for multiple reasons.

First, although a small sample size, our analyses of both an adult and juvenile short-tailed albatross feathers collected from the same region, suggests that both fit in with the pattern we see from archaeological bone, indicating that adults and juveniles can both have isotopic compositions consistent with our dataset.

Second, as we've outlined, the ethnographic data we have at hand, plus knowledge on limitations of how these birds can be hunted, suggests that age bias is an unlikely explanation for the patterns we have observed. This is because historical records indicate that Indigenous communities in the region took both adult and juvenile birds. Moreover, to the extent that harvesting of juveniles could result in an

inflation of isotopic variation, historic records suggesting that juveniles could have been at least as (if not more) abundant than adults in the area prior to their extirpation indicate that ample opportunities to take juveniles would have been available in the past (i.e., this would contribute to greater variation in our sample, which is not what we have observed).

In sum, we have evidence suggesting not only that isotopic variation between age categories is not greater than that seen in our archaeological populations but also that individuals with a wide range of ages were targeted by Indigenous groups locally. Irrespective of differences between how short-tail albatross were selected by Yuquot site residents vs. at other comparative sites, these factors alone suggest that isotopic variation associated with harvesting practises that target birds of different ages is not likely to provide a valid explanation for why so little variation is observed in our sample.

Some minor issues:

6.
 - a. I felt the MS lacks a bit of focus in places, it is set up in a conservation context, but not exactly clear until the very end how this new info informs management etc, I think it could be tighter and a bit more clearly structured in this sense.

We do appreciate that this manuscript involves a good deal of ecological and archaeological context description that may not be germane for all readers. However, given the highly interdisciplinary nature of this project, we feel that all of this information is needed. For instance, information that may seem less relevant for readers in ecology may be deemed essential or conventional for archaeological readers and *vice versa*. Furthermore, much of the contextual detail we've included is critical for setting up the arguments we made and, without this, readers may struggle to see how we were able to arrive at our conclusions. We are therefore reluctant to remove contextual information in an effort to tighten things up.

7. There are some sweeping statements, particularly in the intro that need attention
 - a. e.g. the first, sentence of abstract does not really make sense, it is competition (which of course is a product of population size and resource availability) that is thought to be a major driver of IFSF, along with social facilitation and this it is not well understood.

We fully agree with Reviewer 2 here. The word limit for this abstract is very tight making it difficult to include all the qualifying clauses for statements that we would have liked. In this instance, we were expecting readers to be able to make the link between competition and population pressure. Nonetheless, we have now changed the text to qualify our statement accordingly.

- b. Likewise, the statement on increasing population size and hotspot expansion is just one potential response, there are others.

It is not clear which specific passage Reviewer 2 is referring to here. We did not include a statement suggesting hotspot expansion would be the sole or even primary response to increasing population size. Instead, we simply introduce it as one likely response. Our language here actually introduces this idea with the qualifying clause “, for instance,” signalling this this is only one of a number of possible responses to population growth. To further clarify this, however, we have added text to explicitly note (Line 82) that this is one of many potential responses.

8. Isotopic niche is a well described concept and has been for around 15 years, it does not need inverted commas, just cite a source.

This change has been made. We appreciate that the concept is well established in many areas, but because this may be the first time readers from some areas come across the phrase, we'd opted to highlight it with semi quotes.

9. You need to be careful of contradictory arguments. For example, one of the rationales for the MS is that contemporary understanding of the ecology of this species is biased by the fact that it is a recovering population, but then you use contemporary foraging information to argue that the drivers of variation in stable isotope ratios is about foraging and not space

This is an unavoidable situation in historical ecology, however, we do use historical observations, in addition to modern data, to back up our claims. Moreover, there is no reason that some aspects of modern behaviour can't be used to constrain our understanding of pre modern animal behaviour. A close reading of the manuscript will show that we have mainly used modern examples to show what the species is behaviourally capable of doing so as to contextualize aspects of our archaeological patterns. This use of modern data does not, however, contradict the rationale that historical and archaeological information can inform modern conservation efforts.

10. There are a lot of types e.g. sever v severe line 105

This change has been made.

Reviewer #3 (Remarks to the Author):

This paper reports a detailed stable isotope analysis of bone collagen from archaeological specimens of the Short-tailed albatross recovered from the Yuquot Site on Vancouver Island, B.C. The bone samples range in age over the past 4250 years. Bulk carbon and nitrogen was completed on 95 bones from four distinct time zones within this period to determine if there had been temporal variation in foraging zones over this period. The authors place this study into the context of the modern collapse of this species' populations in the north Pacific. In addition, they compare their results with four other regional studies on this species to show that the Yuquot sample had a distinct foraging area, possibly related to fishing 'hotspots' that once existed near Vancouver Island.

The study is well designed and the paper is well written and thorough in their analyses and discussion. I had only a few suggestions that should be addressed.

1. First, the bone samples analyzed indicate that used a variety of skeletal elements to extract bone collagen, including the tarsometatarsus, carpometacarpus, coracoid, and humerus. No analysis of isotopic results by different skeletal elements was provided and I think it would be helpful to do so, to show that the isotopic signals are the same regardless of element used in the analysis. Since some of these elements do vary in their development and degree of use in life, it would be important to include this analysis here. I also note that the study on the bones from Japan used the same bone (carpometacarpus) for each measurement, probably to avoid any inter-skeletal variation in their results.

This is an excellent observation. We have now added text giving comparisons for this to the Results section (Lines 198-201) and supplementary file. These changes note that mean differences in isotopic compositions between elements are very small (ca. <0.9‰) and would not impact our interpretations. Another key point to be made here is that, while sampling different elements could, in theory, increase the level of variation we'd expect to see, because our arguments are based on the observation of low variation this potential issue has clearly not caused any problems. To offer further clarification, we were not concerned about the potential impact of sampling different elements during the research design phase for this project for a couple additional reasons.

First, while it is true that different bones (or even different parts of the same bone) will have differing turn over rates, the point at which this can potentially become an issue (i.e., when turn over rates differ dramatically) is mainly when comparing more robust (e.g., larger long bones) vs. smaller (e.g., ribs) elements or cortical vs. trabecular bone in animals with dramatic (isotopically) directional diet shifts. In this case, although we have included different bones, we've intentionally targeted bones with similar qualities (denser cortical material from long bones).

Second, as recent research by Hyland, et al. (2021), which we now cite, shows that the typical amount of inter-element/intra-individual isotopic variation observed in animals is actually relatively low (but still higher than observed between means for each bone type here), and is much smaller than the threshold that would be required to have an impact on our interpretations.

All of this said, we did initially aim to sample the same bones as Eda, et al. (2012), which would have allowed us to also use their regression equations to calculate size. Unfortunately, the best MNI (minimum number of individual) counts per context that we could get did not involve using *carpometacarpi*.

2. Second, the authors could have included analysis of sulfur isotopes to further differentiate foraging areas by time period. Sulfur is becoming increasingly useful in this regard and studies have shown that carbon and nitrogen may not vary significantly by time zone or region, but when adding sulfur another dimension in foraging history is revealed. It may be that there was not enough bone collagen from these bird bones to run that analysis along with carbon and nitrogen so I only mention this for future consideration.

This is another interesting point and one that we have closely considered. Sulfur isotope ($\delta^{34}\text{S}$) analyses are becoming increasingly important in a wide range of archaeological and ecological analyses. Unfortunately, owing to very small amount of sulfur in collagen (<1% - just the one atom per methionine residue), $\delta^{34}\text{S}$ analyses require 16x the sample size (8.0+ mg) relative to $\delta^{13}\text{C}$ and $\delta^{15}\text{N}$ (both analyzed on just 0.5mg). Unfortunately, only a small fraction of our samples produced this much materials, meaning that $\delta^{34}\text{S}$ analyses are less feasible for this study.

There are, however, several sound theoretical reasons, rooted in our understanding of marine sulfur cycling/sources, that suggest $\delta^{34}\text{S}$ analyses would be unlikely to provide additional useful information. Firstly, it is well known that the $\delta^{34}\text{S}$ composition of seawater (the main sulfur source for marine ecosystems) is isotopically homogenous across the globe (Canfield 2001; Thode 1991). This means that, by and large, marine animals will have the same $\delta^{34}\text{S}$ value, regardless of where they feed (e.g., Nehlich, et al. 2013). There are a couple of exceptions to this rule, but we do not expect these to apply in the kinds of areas in which short-tailed albatross forage. These exceptions have to do with use of sulfide derived sulfur (having a much lower $\delta^{34}\text{S}$ than sea water), which has been shown to influence the $\delta^{34}\text{S}$ compositions of consumers in select seagrass habitats (e.g., Chasar, et al. 2005; Guiry, et al. 2021) and areas where benthic micro algae (e.g., Szpak and Buckley 2020; Yamanaka, et al. 2003) play a major role. Put another way, and in short, based on the pelagic feeding behaviour of short-tailed albatross and the nearshore nature (i.e., photic zone needs to overlap with ground-fixed primary production) of known sources of marine $\delta^{34}\text{S}$ variation, it's very unlikely that $\delta^{34}\text{S}$ analyses of our samples would add to the picture.

In this context, for the limited cases where we have sufficient remaining collagen from samples, we would prefer to archive these materials for future research that may have more potential. That said, if this is something the reviewer or editors feel strongly about, we are willing to dedicate the residual collagen to adding $\delta^{34}\text{S}$ analyse to this dataset.

3. In addition, regarding the analysis by time zones, did the authors consider the Suess effect for the modern time zone 4? If this effect was negligible in the north Pacific, this should be mentioned in the methods as it could influence their comparisons of that time zone with the earlier periods.

This is another keen point. We have not corrected Zone 4 samples for the Sues effect and we have now included a note about this to the Table 1 caption. This is because it is highly likely that most samples from Zone 4 predate the major Sues effect impacts that began in the late-nineteenth century. The date range for Zone 4 actually terminates the year that the site began to undergo excavation. This was simply a formality as the excavators could not say when the last human activities contributing albatross bones to the site occurred. However, from the earliest ethnographical literature in the region, it is clear that short-tailed albatross collection in the area had ceased by the at least the early twentieth century and possibly much earlier than that. For this reason, it stands to reason that none of the Zone 4 samples require Sues corrections.

4. Third, the discussion is rather long (7.5 pages) with no subheadings to help break up the different topics outlined here. It would be helpful to organize this section a little better using subheadings and avoid some of the repetition within it. I found myself getting a little lost in which topic was most important in their interpretations.

We have now added subheading to better signpost the discussion for readers.

The figures and tables are useful and important for the study.

5. Note, though, that there is a typo in Fig. 1: Jaun de Fuca should be Juan de Fuca.

This has been corrected.

6. The caption for Table 1 should mention that there is no statistical differences in carbon and nitrogen among the time periods.

This information has been added.

7. The caption for Table 2 should indicate which means are statistically different (I think the low mean carbon value from Japan is significantly different from the sample from Oregon, and nitrogen from the sample from California?).

In terms of inter-site comparisons, we only perform statistical analyses oriented towards distinguishing between degrees of isotopic variation. We have opted not to compare inter-site group means because, due to potential overlap between regional baselines, these results would lead to vague interpretations. For instance, although demonstrating a statistical difference between mean isotopic compositions among sites could be useful, failure to demonstrate a statistical difference between groups does not necessarily mean that they did not primarily forage in different regions. In contrast, comparing inter-site group variation does provide a way of demonstrating larger-scale patterns in foraging behaviour. For this reason, we would prefer not to make this change so as to avoid potential confusion among readers. That said, if this is something the reviewer or editors feels strongly about, we will make this change.

References

- Canfield D. 2001. Biogeochemistry of sulfur isotopes. *Reviews in mineralogy and geochemistry* 43(1):607-636.
- Chasar LC, Chanton JP, Koenig CC, Coleman FC. 2005. Evaluating the effect of environmental disturbance on the trophic structure of Florida Bay, U.S.A.: Multiple stable isotope analyses of contemporary and historical specimens. *Limnology and Oceanography* 50(4):1059-1072.
- Eda M, Koike H, Kuro-o M, Mihara S, Hasegawa H, Higuchi H. 2012. Inferring the ancient population structure of the vulnerable albatross *Phoebastria albatrus*, combining ancient DNA, stable isotope, and morphometric analyses of archaeological samples. *Conservation Genetics* 13(1):143-151.
- Guiry E. 2019. Complexities of stable carbon and nitrogen isotope biogeochemistry in ancient freshwater ecosystems: Implications for the study of past subsistence and environmental change. *Frontiers in Ecology and Evolution* 7(313).
- Guiry EJ, Kennedy JR, O'Connell MT, Gray DR, Grant C, Szpak P. 2021. Early evidence for historical overfishing in the Gulf of Mexico. *Science Advances* 7(32):eabh2525.
- Hedges REM, Clement JG, Thomas C, L. D, O'Connell TC. 2007. Collagen turnover in the adult femoral mid-shaft: Modeled from anthropogenic radiocarbon tracer measurements. *American Journal of Physical Anthropology* 133(2):808-816.
- Hobson KA, Clark RG. 1992. Assessing avian diets Using stable isotopes I: Turnover of ^{13}C in tissues. *The Condor* 94(1):181-188.
- Hyland C, Scott MB, Routledge J, Szpak P. 2021. Stable Carbon and Nitrogen Isotope Variability of Bone Collagen to Determine the Number of Isotopically Distinct Specimens. *Journal of Archaeological Method and Theory*.
- Laws EA, Popp BN, Bidigare RR, Kennicutt MC, Macko SA. 1995. Dependence of phytoplankton carbon isotopic composition on growth rate and $[\text{CO}_2]_{\text{aq}}$: Theoretical considerations and experimental results. *Geochimica et cosmochimica acta* 59(6):1131-1138.
- Nehlich O, Barrett JH, Richards MP. 2013. Spatial variability in sulphur isotope values of archaeological and modern cod (*Gadus morhua*). *Rapid Communications in Mass Spectrometry* 27(20):2255-2262.
- Popp BN, Laws EA, Bidigare RR, Dore JE, Hanson KL, Wakeham SG. 1998. Effect of phytoplankton cell geometry on carbon isotopic fractionation. *Geochimica et cosmochimica acta* 62(1):69-77.
- Rau GH, Takahashi T, Des Marais DJ. 1989. Latitudinal variations in plankton $\delta^{13}\text{C}$: implications for CO_2 and productivity in past oceans. *Nature* 341(6242):516.
- Sigman D, Karsh K, Casciotti K. 2009. Ocean process tracers: Nitrogen isotopes in the ocean. In: Steele JH, Turekian KK, Thorpe SA, editors. *Encyclopedia of Ocean Science*. 2 ed. San Diego: Academic Press. p 4139-4152.
- Szpak P, Buckley M. 2020. Sulfur isotopes ($\delta^{34}\text{S}$) in Arctic marine mammals: indicators of benthic vs. pelagic foraging. *Marine Ecology Progress Series* 653:205-216.
- Thode H. 1991. Sulphur isotopes in nature and the environment: an overview. In: HR K, VA G, editors. *Stable isotopes: natural and anthropogenic sulphur in the environment*. Chichester: Wiley. p 1-26.
- Vokhshoori NL, McCarthy MD, Collins PW, Etnier MA, Rick T, Eda M, Beck J, Newsome SD. 2019. Broader foraging range of ancient short-tailed albatross populations into California coastal waters based on bulk tissue and amino acid isotope analysis. *Marine Ecology Progress Series* 610:1-13.
- Yamanaka T, Mizota C, Shimoyama S. 2003. Sulfur isotopic variations in soft tissues of five benthic animals from the reductive, tidal-flat sediments in northern Kyushu, Japan. *Marine Biology* 142(2):327-331.

REVIEWERS' COMMENTS:

Reviewer #1 (Remarks to the Author):

In the "response to reviewers" file, the authors provided thorough, convincing responses to all reviewers comments, concerns and made appropriate changes to the manuscript. I do not have any further requests for changes.

Reviewer #3 (Remarks to the Author):

My review comments have been addressed. I would only add that because two reviewers (including me) questioned why Table 2 lacked statistical comparisons, a brief explanation as provided in the rebuttal should be in the methods of the paper (or in the table caption) as I'm sure readers will continue to question this. So, it would help to explain up front the lack of statistical analysis of these data.